# Oxidative Stress Markers and Histopathological Changes in Selected Organs of Mice Infected with Murine Norovirus 1 (MNV-1)

**DOI:** 10.3390/ijms25073614

**Published:** 2024-03-23

**Authors:** Paulina Janicka, Dominika Stygar, Elżbieta Chełmecka, Piotr Kuropka, Arkadiusz Miążek, Aleksandra Studzińska, Aleksandra Pogorzelska, Katarzyna Pala, Barbara Bażanów

**Affiliations:** 1Department of Pathology, Wrocław University of Environmental and Life Sciences, Norwida 31, 50-375 Wrocław, Poland; paulina.janicka@upwr.edu.pl (P.J.); aleksandra.pogorzelska@upwr.edu.pl (A.P.); 2Food4Future Technologies Sp. z o.o., Tarasa Szewczenki 24, 51-351 Wroclaw, Poland; kasia@f4ftech.com; 3Department of Physiology, Faculty of Medical Sciences in Zabrze, Medical University of Silesia, Jordana 19, 41-808 Zabrze, Poland; dstygar@sum.edu.pl; 4Department of Statistics, Department of Instrumental Analysis, Faculty of Pharmaceutical Sciences in Sosnowiec, Medical University of Silesia in Katowice, Ostrogórska 30, 41-200 Sosnowiec, Poland; echelmecka@sum.edu.pl; 5Division of Histology and Embryology, Department of Biostructure and Animal Physiology, Faculty of Veterinary Medicine, Wrocław University of Environmental and Life Sciences, Norwida 25, 50-635 Wroclaw, Poland; piotr.kuropka@upwr.edu.pl; 6Department of Biochemistry and Molecular Biology, Wroclaw University of Environmental and Life Sciences, Norwida 31, 50-375 Wroclaw, Poland; arkadiusz.miazek@upwr.edu.pl (A.M.); aleksandra.studzinska@upwr.edu.pl (A.S.)

**Keywords:** oxidative stress, bacterial and viral diseases

## Abstract

This paper describes the effects of murine norovirus (MNV) infection on oxidative stress and histopathological changes in mice. This study uses histopathological assays, enzymatic and non-enzymatic antioxidant markers, and total oxidative status and capacity (TOS, TAC). The results suggest that MNV infection can lead to significant changes with respect to the above-mentioned parameters in various organs. Specifically, reduced superoxide dismutase (SOD), Mn superoxide dismutase (MnSOD), catalase (CAT), and glutathione reductase (GR) activities were observed in liver tissues, while higher MnSOD activity was observed in kidney tissues of MNV-infected mice when compared to the control. GR activity was lower in all tissues of MNV-infected mice tested, with the exception of lung tissue. This study also showed that norovirus infection led to increased TOS levels in the brain and liver and TAC levels in the brain, while TOS levels were significantly reduced in the kidneys. These changes may be due to the production of reactive oxygen species (ROS) caused by the viral infection. ROS can damage cells and contribute to oxidative stress. These studies help us to understand the pathogenesis of MNV infection and its potential effects on oxidative stress and histopathological changes in mice, and pave the way for further studies of the long-term effects of MNV infection.

## 1. Introduction

Viral infections are one of the major causes of global health issues [1], where noroviruses (NoVs) are one of the most common causes of gastroenteritis in humans in all age groups worldwide [2]. These pathogens belong to the family *Caliciviridae* and are a type of non-enveloped, single-stranded RNA viruses characterized by being positive sense [3]. Noroviruses have also been identified in a variety of animal species, including pigs, cows, sheep, cats, dogs, rats, and mice, but murine norovirus (MNV) is the only representative of this family showing the ability to replicate in cell culture and laboratory animals [4], and at the same time, it is one of the most common pathogens of laboratory mice [5].

The initiation of effective therapy, as well as ensuring the removal of side effects caused by viral infections, requires a deep understanding of the pathogenesis and effects induced by norovirus infections at the cellular, tissue, and organ levels.

Wild-type mice (adults) infected with MNV show subclinical intestinal infection without gastric symptoms [6]. In contrast, MNV infection is lethal in immunocompromised mice (for example in immunodeficient mice with impaired innate immunity and negative expression of STAT1 and *Rag2* genes). There has been a steady increase in MNV infections in laboratories, where non-susceptible mouse strains infect susceptible mice. After administration of the virus to the immunocompromised animals, systemic disease develops—bloating, intestinal pathology, and diarrhea that result in weight loss [7,8,9]. Currently, MNV is the most widespread pathogen in animal facilities [10]. Mice infected with asymptomatic MNV are the source of the virus, and they can adversely affect other laboratory animals and undermine the reliability of studies conducted on animals [11]. Several histopathological changes were observed during MNV infection in different strains of laboratory mice. NOD.CB17-Prkdcscid/J (NOD-scid) immunodeficient mice showed no lesions, while liver and small intestine lesions were noted in STAT1-/- mice, and *Rag2*/STAT1-/- mice showed liver, lung, and brain lesions. The latter were also seen in infected animals of the IFNγR-/- strain [4,12,13]. Histopathological changes in the colon were observed in immunocompromised B6.129P2-Il10tm1Cgn/JZtm, C3Bir.129P2-Il10tm1Cgn/JZtm, and germ-free B6.129P2-Il10tm1Cgn/JZtm mice [11].

Pathological changes in organs observed over the course of infection may result from the disruption of metabolic pathways within the infected cell. Viral proliferation takes over the functions of host cells and causes a significant imbalance in the intracellular physiological processes and systems, including the redox system [10]. Oxidative stress, resulting from the imbalance between the antioxidant systems and toxic reactive oxygen species (ROS) [14], can be triggered by a wide variety of viral infections, including HIV 1, hepatitis B, C, and D viruses, herpes viruses, and respiratory viruses, such as coronaviruses [15]. However, oxidative stress and ROS generation are essential for numerous biological and physiological functions. Increased ROS concentration triggers enzymatic antioxidant systems, such as superoxide dismutase (SOD, MnSOD), glutathione peroxidase (GPx), glutathione reductase (GR), and catalase (CAT), considered the main oxidative stress markers, and non-enzymatic antioxidant systems including total oxidative status (TOS), total antioxidant capacity (TAC), and malondialdehyde (MDA) [16]. 

Superoxide dismutase (SOD), with its isoforms, e.g., MnSOD, is one of the main antioxidative enzymes that neutralize superoxide ions [17]. It mainly catalyzes the conversion of superoxide radicals (O2•−) to hydrogen peroxide (H_2_O_2_), which is subsequently converted into H_2_O by GPx and CAT [18]. Oxidative stress caused by an imbalance in oxidant/antioxidant markers and disruption of endogenous antioxidant systems can be quantified using total antioxidant capacity (TAC), total oxidative status (TOS) [19], and MDA concentration, which is considered a marker of oxidative stress, more specifically of lipid peroxidation understood as lipid damage caused by ROS [20]. 

The comprehension of relevant antioxidant mechanisms and the relationship between viral infection pathological changes and oxidative stress may play a crucial role in understanding viral pathogenesis and identifying potential therapeutic targets, especially as antiviral therapies are often unavailable. Therefore, instead of acting directly on the virus, the effects of infection can be reduced by using therapy to limit the effects of oxidative stress.

The purpose of this project was to study the effects of MNV infection on oxidative stress and histopathological changes in C56Bl/6J mice.

## 2. Results

### 2.1. Assessing the Effectiveness of Infection

Due to the rapid rate of elimination of the virus from the body, RT-PCR tests showed such low titers that the result was not statistically significant. Successful infection of mice was confirmed by the detection of antibodies to norovirus in the serum of infected mice by the virus neutralization test (VNT) and interferon measurements. 

### 2.2. Monitoring Histopathological Changes in Organs of MNV-Infected Mice

Despite the fact that tissue sections were taken from the small and large intestine, pancreas, liver, heart, lungs, kidneys, and brain fragments (forebrain, midbrain) after 3, 4, and 7 days of norovirus incubation, similar changes were found in the liver and brain in all individuals studied. In the other organs, there were either no changes (small and large intestine, pancreas, heart) or individual changes independently of the length of virus incubation (lungs, kidneys). 

In the lung and kidney, small perivascular infiltrates within the blood vessels compared to a normal picture were observed. As infiltrates were found in the subcapsular zone as well as in the cortex and medulla of the kidney and in the lungs surrounding the bronchioles in the control group, it was decided not to consider changes in these organs as caused by the virus (Figure 1).

In the liver, perivascular infiltrates were present surrounding both the hepatic triads and intralobular vessels. In the lobules, they often surrounded single hepatocytes; however, no features of necrosis or apoptosis were found. In the liver, numerous binucleated hepatocytes were present in all experimental groups, which, however, indicated enhanced repair processes (Figure 2).

In the brain, infiltrates were localized in the meningeal vessels within the cerebellum and midbrain, mostly in white matter and pia mater. No such infiltrates were found in the forebrain. The infiltrates were often accompanied by edema, giving the white matter a foam-like character (Figure 3 and Figure 4).

At 4 and 7 days after infection, the changes were broadly similar. Although there was less swelling in the cerebellum 7 days after infection compared to 3 days after infection, there were also lower levels of blood present in the vessels in the organs. In addition, an increased number of small infiltrates was observed in the liver and brain compared to 3 days post-infection. Nevertheless, no increased apoptosis or necrosis was observed in the organs examined (Figure 1, Figure 2, Figure 3 and Figure 4).

### 2.3. Oxidative Stress Markers

We found statistically significant differences in SOD, MnSOD, GST, GPx, GR, and CAT activities, TAC and TOS levels, and MDA concentration between the sampled tissues in the control and norovirus-infected mice (Table 1).

We found higher SOD activity in the liver tissue of control mice when comparing it in individual tissues between the control and norovirus-infected mice (Table 1). Similarly, we found statistically significant differences in MnSOD activity when comparing it in individual tissues between the control and norovirus-infected mice. We observed higher MnSOD activity in the liver, but lower MnSOD activity in the kidney tissues of the control mice (Table 1). Further comparisons within individual tissues showed higher GR concentrations in the brain, cerebellum, and liver tissues of the norovirus-infected mice compared to control mice. We observed higher GR activity in the lung and kidney tissues of the control mice compared to norovirus-infected mice (Table 1). Comparisons within individual tissues showed higher CAT activity in the cerebellum, lung, and kidney tissues of the norovirus-infected mice compared to control mice. We observed increased CAT activity in the liver of the control mice compared to the norovirus-infected ones. (Table 1).

When comparing GPx activity in individual tissues between control and norovirus-infected animals (Table 1), we found higher GPx activity in the cerebellum and lungs of control mice, as well as in the liver tissue of norovirus-infected mice. Additionally, this comparison revealed higher GST activity in the liver of norovirus-infected mice (Table 1). Furthermore, we observed higher TAC levels in brain tissue but lower levels in the cerebellum and liver tissues of norovirus-infected mice compared to control mice. Conversely, we noted higher TOS levels in the brain and liver, but lower TOS levels in the kidney tissues of norovirus-infected mice compared to controls. Finally, concentrations of MDA were higher in the liver tissue of control mice compared to norovirus-infected mice.

## 3. Discussion

The present study was designed to investigate how MNV infection affects oxidative stress and histological changes in mice.

It is known that MNV infection is associated with histopathological changes in immunocompetent hosts, but clinical disease is prevented by a STAT1-dependent interferon response [21]. Interferon gamma, along with other type I and III interferons, has previously been shown to play a role in limiting MNV macrophage infection [22].

We and others have used several inbred strains of mice as hosts for MNV infection [1,23,24]. But, since most of the genetically engineered mouse strains are from the C57Bl/6 genetic background, in this report, we decided to use this host strain. Moreover, a direct comparison of MNV-associated diarrhea in C57Bl/6 and Balb/c mice showed that both strains are comparably susceptible [25]. In our project, we also considered the use of different MNV strains. In studies of the pathogenesis of this virus in mice, the authors usually use MNV-1.CW 1, MNV-1.CW 3, and MNV-4, -5, and -6. These strains induce histopathological changes in many organs in some experiments, while in other studies, these changes are weak or not noted at all. This depends on many factors—the immunocompetence of the animals, their age, and, finally, the dose of virus [26]. Mumphrey et al. [27], comparing MNV-1.CW 3 and CW 1, reported that the MNV-1.CW 3 strain, being more virulent than CW 1, caused more severe histopathological changes. However, these studies, as well as those of other authors [5], do not apply to the neural tissue in which we wanted to demonstrate lesions. In a publication by Karst et al. [5], such changes were obtained using purified MNV-1 from IFNαβγR-/mouse brains. To obtain similar changes, we chose the MNV-1.CW1 (ATCC) strain of the same origin. Here. we also analyzed oxidative stress markers, which were tissue dependent, suggesting variation in oxidative stress associated with viral pathogenesis.

In the study by Hsu et al. [28], MNV-infected mice were analyzed for histopathological changes in various groups, including naturally MNV-infected mice; wild-type mice experimentally infected with MNV; immunodeficient mice experimentally infected with MNV; mice in gastrointestinal disease models experimentally infected with MNV; and mice in other disease models experimentally infected with MNV. Some naturally MNV-infected mice showed histopathological changes in the liver, lungs, mesenteric lymph nodes, lungs, and spleen. In some wild-type mice experimentally infected with MNV, histopathological changes occurred in the liver, small intestines, and spleen. In the histopathological examination of immunodeficient mice experimentally infected with MNV, changes were observed in the liver, lungs, small intestine, and spleen, and in the case of *Rag2*-/-, /STAT1-/-, and IfnαβγR-/- mice, changes were observed in the brain. Some mice in gastrointestinal disease models experimentally infected with MNV exhibited histopathological changes in mesenteric lymph nodes, small and large intestines, the spleen, and the stomach. In the case of some mice in other disease models experimentally infected with MNV, changes were observed in mesenteric lymph nodes and the aorta [29]. In our experiment, no severe changes in examined organs were found except mild lymphocyte infiltrations in the brain and liver without visible cell damage [21].

In fact, some of the most common causes of encephalitis are viral infections. Several viral agents have been described to cause this condition, such as arboviruses, rhabdoviruses, enteroviruses, herpesviruses, retroviruses, orthomyxoviruses, orthopneumoviruses, and coronaviruses, among others. However, MNV was not treated as an agent that can cause encephalitis [30]. 

The present study also showed that the enzymatic and non-enzymatic markers of oxidative stress were tissue dependent. Viral infection caused depletion of most of the assessed enzymes like SOD, MnSOD, CAT, and GR in liver tissue compared to the control mice, whereas the GPx and GST activities measured in liver tissue were significantly higher in the norovirus-infected mice. Superoxide dismutase, catalase, and glutathione-dependent enzymes, such as GPx, are the most important antioxidant enzymes. Superoxide dismutases, like total SOD and MnSOD, are enzymes that catalyze the conversion of superoxide into O_2_ and H_2_O_2_. Catalase catalyzes the conversion of H_2_O_2_ into H_2_O and O_2_, and GPxs catalyze the conversion of H_2_O_2_ or hydroperoxides to H_2_O and alcohols while oxidizing GSH to GSSG. Reactive oxygen species’ role during viral infections may be unclear and may depend on how they are produced [31]. The activity of MnSOD was higher in the kidney tissue of the norovirus-infected mice when compared to control ones. The activity of GR was lower in all the studied tissues in norovirus-infected mice except for lung tissue. It is known that the activities and concentration of the selected antioxidative enzymes change during viral infection or different stages of infection. A study conducted on an in vitro model reported that during the first hours after infection, SOD, GST, CAT, and GPx were the main antioxidative enzymes induced in response to the infection [32]. As the infection progressed, only SOD concentration increased, leading to increased H_2_O_2_ production, whereas other antioxidant enzymes decreased, including those that are critical for neutralizing H_2_O_2_ [33].

The body’s overall oxidative state is typically estimated using the total oxidant status (TOS), whereas the body’s overall antioxidant level is determined by total antioxidant capacity (TAC) [34]. Other studies showed that TOS levels clearly increase with the aggravation of the COVID-19 disease, since the elevated level of oxidative stress has the capability to intensify the severity of COVID-19. Karkhanei et al. reported that serum TOS level, one of the oxidative stress biomarkers, was higher in patients with acute SARS-CoV-2 presentation. The authors showed that serum TOS levels in infected COVID-19 patients were related to other important factors, such as fever, hospitalization longer than one week, residency, educational status, and job type, and indirectly related to SpO2 (oxygen saturation). In the present study, the selected non-enzymatic antioxidative and oxidative parameters such as TOS, TAC, and MDA changed after the norovirus infection, depending on the analyzed tissue. Nevertheless, noroviral infection resulted in increased TOS levels in the brain and liver and increased TAC levels in brain tissues, while TOS levels were significantly decreased in the kidneys. The decrease in TAC may lead to reduced body resilience and ultimately to increased mortality [35]. The presented results prove that oxidative stress markers change rapidly depending on the conditions of the external environment. Selected evidence suggests that ROS overproduction and antioxidant system depletion play a significant role in the pathogenesis of viral infections and in the progression of the disease severity.

With high levels of oxidative stress in target organs including the brain, liver, kidney, and lung, oxidative stress plays a crucial role in a number of pathological disorders. We believe that this shows a strong connection between oxidative stress levels and the specific functions of each organ. Multiple intracellular signaling pathways are known to be activated by oxidative stress, resulting in subsequent organ failure. Targeting oxidative stress is thus thought to be useful in preventing organ damage, and assessing the level of oxidative stress may act as a biomarker in a variety of disease conditions. Here, we understand that the selected virus may have varied effects and, thus, create varying stresses in different organs—depending on the location of inflammation, the severity of the infection, and the corresponding immune response—which may also be influenced by other factors. In order to support this hypothesis further, biomolecular studies have to be performed.

## 4. Materials and Methods

### 4.1. Mice

Animal experimentation was carried out under permission no. 022/2022 from the Local Ethical Committee in Wroclaw, Poland. Mice were housed in individually ventilated cages and fed with standard sterilized rodent chow and water ad libitum. The specific pathogen-free, hygienic status of the animal facility is maintained by periodic sentinel screening according to FELASA recommendations. All measures were taken to minimize the number of animals in procedures and their suffering. 

The study material consisted of healthy and norovirus-infected mice, which were divided into 4 study groups: Healthy controls (control);Euthanized 3 days after infection (3 days);Euthanized 4 days after infection (4 days);Euthanized 7 days after infection (7 days).

The method used to euthanize mice was cervical dislocation. Before the procedure, the mice were placed in the induction chamber of a veterinary anesthesia workstation (Équipement Vétérinaire Minerve, Esternay, France). The isoflurane flow was set to 2.5%. Two minutes after the mice were fully premedicated, they were transferred to pads with an inhalation mask and the anesthetic administration was continued. With a firm grip, placing the index finger and thumb directly behind the skull, and with the other hand grabbing the tail just at the base, one strong movement led to the dislocation of the cervical vertebrae. Pulling was continued until the skull was felt to be completely unattached to the spine. Each time, breathing stopped within a few seconds of the dislocation. Before starting the necropsy procedure, mice were decapitated. 

### 4.2. Monitoring the Kinetics of MNV-1 Infection in C57Bl/6J Mice

The amount of virus in the suspension to be administered to the experimental animals was tested by RT-PCR. C57Bl/6J mice were infected per os with 50 microliters of MNV1 suspension (100TCID_50_) on day 0. Mice were kept in isolation under specific pathogen-free (SPF) conditions to prevent accidental infection with another pathogen. After MNV was experimentally administered to the animals, a repeat RT-PCR test was performed after euthanasia. We used an indirect interferon gamma (IFNγ) test and a virus neutralization test to ascertain comparable MNV infection and antibody levels between individual mice.

### 4.3. Cell Culture

In vitro experiments were conducted using RAW 264.7 macrophage cells (TIB-7 TM, ATCC, Manassas, VA, USA). These cells were cultured in Dulbecco’s Modified Eagle’s Medium (DMEM) supplemented with nonessential amino acids and 10% fetal bovine serum (FBS), obtained from Biological Industries (Kibbutz Beit-Haemek, Israel). The cell cultures were maintained in 25 mL polystyrene flasks (Googlab Scientific, Rokocin, Poland) and incubated at 37 °C in a controlled environment with 5% CO_2_ and 95% humidity. Passage of the cells was carried out using 0.25% trypsin–EDTA (Biological Industries).

### 4.4. Virus Propagation

Murine norovirus (ATCC, Los Angeles, CA, USA, VR-1973) was used for this project. When the RAW 264.7 (TIB-7^TM^, ATCC, Manassas, VA, USA) cells reached 75–80% confluence, the cell culture medium was removed, and the cells were rinsed with phosphate-buffered saline (PBS). Subsequently, the virus was introduced into the flask and allowed to incubate for 3 h under conditions of 37 °C with 5% CO_2_. Following this incubation, the virus suspension was removed, and the cell culture was washed again with PBS. Dulbecco’s Modified Eagle’s Medium (DMEM) was then added, and the cells were subjected to further incubation for a period of 4 to 6 days. Daily observations were carried out using an inverted microscope (Olympus Corp., Hamburg, Germany; Axio Observer, Carl Zeiss MicroImaging GmbH, Munich, Germany) to monitor the development of cytopathic effects (CPEs). Norovirus was titrated using a TCID_50_ (50% Tissue Culture Infectious Dose) assay and then stored in a freezer at −80 °C

### 4.5. Histological Examination

For histopathological evaluation, small and large intestine, pancreas, liver, heart, lung, kidney, and brain fragments (forebrain, midbrain) were taken from all animals. The material was fixed in a solution of 4% buffered formalin (pH 7.2–7.4) for 48 h and then dehydrated in an alcoholic series after washing in tap water. The material was then embedded in paraffin. The 6 µm thick sections were stained with hematoxylin and eosin. The material was analyzed in a Nikon Eclipse 80i light microscope using NisElements Ar software https://www.microscope.healthcare.nikon.com/products/software/nis-elements/nis-elements-advanced-research, accesed on 1 June 2023 (Nikon, Tokyo, Japan).

### 4.6. Oxidative Stress

Seven days after viral inoculation, we harvested 100 mg of selected tissues: brain, cerebellum, liver, lungs, and kidneys from the infected animals (n = 7) and put it in 1 mL of a homogenizing buffer with protease inhibitors. The tissues were homogenized (1:10 *w*/*v*) in 0.9% NaCl with a glass homogenizer (Potter-Elvehjem PTFE, Sigma-Aldrich, Darmstadt, Germany) and then sonicated (Virsonic 100, VirTis, Gardiner, NY, USA). The tissue samples were then frozen and stored at −80 °C, until the analysis.

For this study, organs were taken from all control (n = 7) and infected mice were euthanized 7 days after inoculation (n = 7). 

#### 4.6.1. Superoxide Dismutase (SOD) (EC 1.15.1.1) Activity

Total SOD activity was measured using the Oyanagui method [35]. In this method, xanthine oxidase catalyzes the production of superoxide anion that reacts with hydroxylamine to produce nitroso ions. The latter combined with n-(1-naphthyl)ethylenediamine and sulfanilic acid gives a color combination that can be measured spectrophotometrically. The SOD and mnSOD activities were determined using the Oyanagui method [35]. The SOD activities were expressed in NU/mg of protein, with 1 NU (nitrate unit) equal to 50% inhibition of nitrite ion formation.

#### 4.6.2. Catalase (CAT) Activity (EC 1.11.1.6)

CAT activity was assessed using the Aebi method [36]. In this method, the homogenate is mixed with perhydrol in 50 mM TRIS/HCl buffer, pH 7.4, and the reaction is started by adding freshly prepared hydrogen peroxide. The rate of decomposition of hydrogen peroxide can be measured spectrophotometrically at 240 nm. CAT activity is expressed as units per 1 g protein (IU/g protein).

The CAT activity was determined using the Aebi method [36] and expressed as units per 1 g protein (IU/g protein).

#### 4.6.3. Glutathione Peroxidase (GPx) Activity (EC 1.11.1.9)

GPx activity was measured by using the kinetic method [37], with t-butyl peroxide as a substrate. In this reaction, oxidized glutathione (GSSG) is regenerated in the presence of glutathione reductase (GR) and NADPH. GPx activity was expressed as μmoles of NADPH oxidized in 1 min per 1 g of protein (IU/g protein) for selected tissues.

The GPx activity was measured using the kinetic method [37] with t-butyl peroxide as a substrate and is expressed as μmoles of oxidized NADPH in 1 min per 1 g of protein (IU/g protein).

#### 4.6.4. Glutathione Reductase (GR) Activity (EC 1.8.1.7)

GR activity was determined by using the kinetic method and is expressed as μmoles of nicotinamide adenine dinucleotide phosphate (NADPH) utilized in 1 min per 1 g of protein (IU/g protein) for assessed samples [38]. This method is based on changes in the concentration of NADPH that reacts with oxidized glutathione. The changes in absorbance at 340 nm were measured with a PERKIN ELMER Victor X3 reader (PerkinElmer, Inc., Waltham, MA, USA).

The GR activity was determined using the kinetic method [38] and expressed in μmoles of NADPH utilized in 1 min by 1 g of protein (IU/g protein).

#### 4.6.5. Glutathione-S Transferase (GST) Activity (EC 2.5.1.18) 

GST activity was estimated using the Habig and Jakoby kinetic method [39]. The reaction mixture containing reduced glutathione was added to the samples. After initial stabilization, 1-chloro-2,3-dinitrobenzene (in ethyl alcohol solution) was added, and absorbance changes were monitored using a PERKIN ELMER Victor X3 reader, at 340 nm wavelength, for at least 3 min. GST activity was expressed as μmoles of thioether formed within 1 min per 1 g of protein (IU/g protein).

#### 4.6.6. Total Oxidant Status (TOS) and Total Antioxidant Capacity (TAC)

The total oxidant status (TOS) and total antioxidant capacity (TAC) levels were determined using the Erel methods [40,41]. TOS was determined using the methodology provided by Erel based on a system containing xylene orange, o-dianisidine, and Fe^+2^ ions [35]. The determination of TAC in serum was based on the method of decolorization of oxidized ABTS (2,2-azinobis(3-ethylbenzothiazo-ine-6-sulfonate)) under the influence of antioxidants occurring in the tested material [35].

#### 4.6.7. Malondialdehyde (MDA) Concentration

The malondialdehyde (MDA) concentration was measured using a reaction with thiobarbituric acid [42], calculated from a standard curve prepared with 1,1,3,3-tetraethoxypropane, and expressed in μmol/g protein.

#### 4.6.8. Statistical Analysis of Antioxidant Stress Markers

The normality of distributions was assessed using the Shapiro–Wilk test. Kruskal–Wallis ANOVA was used for comparisons in the study and norovirus groups depending on the type of tissue. Within a single tissue, comparisons between the study group and the control group were made using the Mann–Whitney U test. The median (lower–upper quartile) (Me (Q1–Q3)) was used in the description. The tests performed were two-sided and the significance level was set at 0.05. The calculations were performed in the TIBCO Statistica^®^ 13.3.0 program (data analysis software system), (TIBCO Software Inc., Santa Clara, CA, USA), accessed on 15 September 2023 

## Figures and Tables

**Figure 1 ijms-25-03614-f001:**
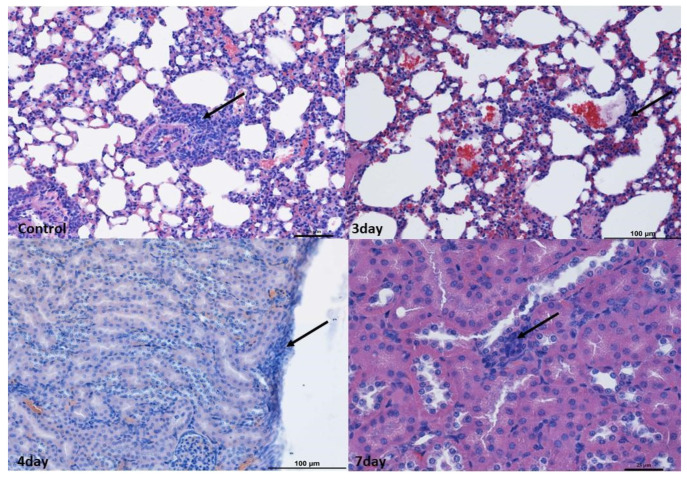
Histopathological changes in the lungs and kidneys. Lungs: Example pictures of lymphocyte infiltration in the control group and on the 3rd day. Note the presence of lymphoreticular tissue present (arrow) next to the bronchiole in the control group as well as in the alveoli on the 3rd day. Similar changes were found in individuals from other groups. Mag 100×. Kidneys: Example pictures of lymphocyte infiltration in animals on the 3rd, 4th, and 7th days. Mild lymphocyte infiltration (arrow) under the kidney capsule and between proximal tubules on the 4th day. Similar changes were observed in other time periods and in the control group. Fourth day—Mag 200×, seventh—Mag 400×. Scale bar—control and 7th day—25 µm, scale bar—3rd and 4th days—100 µm. These changes were considered not to be associated with norovirus.

**Figure 2 ijms-25-03614-f002:**
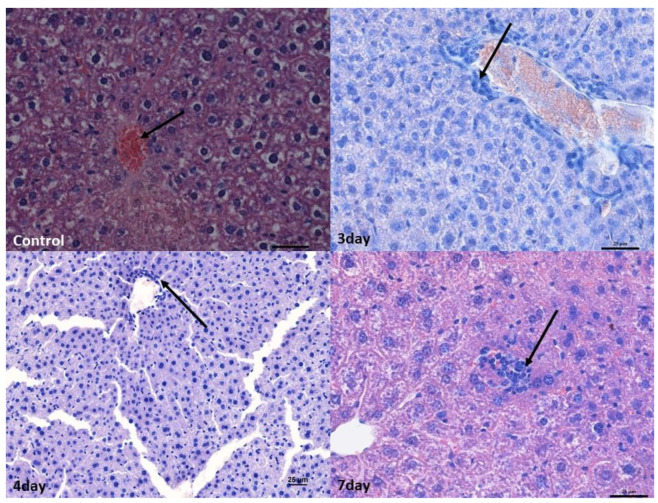
Histopathological changes in the liver. Control group—normal structure of hepatocytes surrounding the central vein (arrow). Mag 400×. Mild lymphocyte infiltration around the blood vessel (arrow) after 3rd (Mag 400×) and 4th days. ((Mag 200×).After the 7th day, lymphocyte infiltration in the hepatic stroma (arrow). Mag 400×. Scale bar—25 µm.

**Figure 3 ijms-25-03614-f003:**
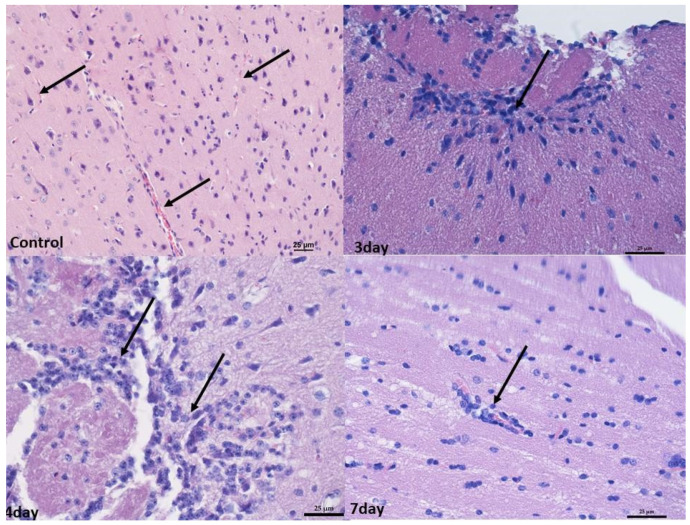
Histopathological changes in the midbrain. Control group—numerous blood vessels in the midbrain (arrow). Mag 200×. After 3rd and 4th days—perivascular infiltration of lymphocytes in the white matter (arrow). Mag 400×. (D) after the 7th day, mild perivascular lymphocyte infiltration in white matter (arrow). Mag 400×. Scale bar—25 µm.

**Figure 4 ijms-25-03614-f004:**
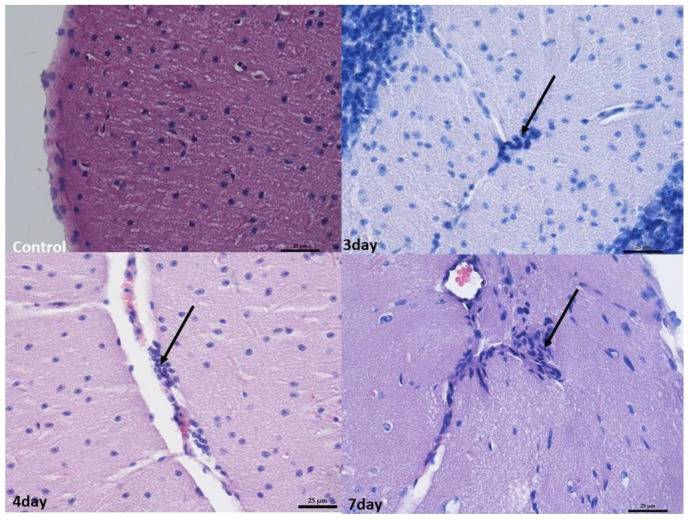
Histopathological changes in the cerebellum. Control group—normal image of unaltered cortex covered with pia matter. Mag 400×. On the 3rd day, 4th day, and 7th day—perivascular infiltration of lymphocytes in the white matter (arrow). Mag 400×. Scale bar—25 µm.

**Table 1 ijms-25-03614-t001:** Levels of oxidative stress markers in the brain, cerebellum, liver, lung, and kidney tissues collected from all control (n = 7) and norovirus-infected mice euthanized 7 days after inoculation (n = 7). Results are presented as median with lower and upper quartiles (Me (Q1–Q3)).

Oxidative Stress Marker	Group	Brain	Cerebellum	Liver	Lungs	Kidney	*p*
**SOD** **(NU/mg protein)**	control	11.1 (9.0–14.1)	15.0 (11.6–18.9)	3.6 (3.3–4.8)	4.1 (2.5–5.7)	5.4 (4.8–6.8)	<0.001
	norovirus	**12.0** (11.2–13.4)	15.0 (11.5–15.5)	**2.7** (2.4–2.8)	**4.9** (3.2–6.3)	**5.9** (5.0–6.1)	<0.001
	p_control vs. norovirus_	0.443	0.443	**<0.01**	0.523	0.701	
**MnSOD** **(NU/mg protein)**	control	3.1 (2.1–4.1)	2.5 (1.4–3.7)	2.4 (2.2–2.9)	2.6 (2.3–3.0)	1.9 (1.8–2.0)	<0.05
	norovirus	**2.7** (2.6–3.0)	**2.6** (2.3–2.8)	**2.0** (1.9–2.2)	2.6 (2.2–2.8)	**2.2** (2.1–2.4)	<0.01
	p_control vs. norovirus_	0.609	0.609	**<0.05**	0.701	**<0.05**	
**GR** **(IU/g protein)**	control	26.6 (25.7–28.0)	26.8 (24.9–27.5)	5.7 (5.4–7.4)	13.5 (12.4–15.9)	1.8 (1.5–2.4)	<0.001
	norovirus	**15.4** (15.0–17.8)	**21.9** (19.3–22.6)	**0.3** (0.2–0.3)	**18.3** (17.8–18.4)	**3.6** (2.6–4.7)	<0.001
	p_control vs. norovirus_	**<0.01**	**<0.01**	**<0.01**	**<0.01**	**<0.05**	
**CAT** **(IU/g protein)**	control	184.1 (141.2–187.7)	133.2 (108.1–147.1)	396.4 (382.7–437.5)	140.3 (133.4–144.3)	290.8 (264.3–298.1)	<0.001
	norovirus	**154.0** (117.6–179.8)	**193.5** (186.3–238.9)	**225.6** (204.1–247.3)	**180.8** (166.1–187.7)	**358.2** (355.0–374.5)	<0.001
	p_control vs. norovirus_	0.250	**<0.01**	**<0.01**	**<0.01**	**<0.01**	
**GPx** **(IU/g protein)**	control	446 (257–555)	488 (413–522)	65 (60–69)	496 (485–530)	270 (208–401)	<0.001
	norovirus	**309** (271–358)	**256** (205–373)	**135** (113–150)	**389** (384–399)	**250** (245–270)	<0.001
	p_control vs. norovirus_	0.201	**<0.01**	**<0.01**	**<0.01**	0.443	
**GST** **(IU/g protein)**	control	6.4 (6.2–7.0)	8.1 (6.8–8.8)	3.9 (3.1–4.6)	6.2 (5.2–7.6)	5.6 (5.2–6.6)	<0.001
	norovirus	**6.5** (5.4–6.8)	**7.5** (6.4–9.6)	**5.9** (4.7–7.5)	**6.4** (8.9–7.0)	**5.1** (5.1–5.2)	<0.05
	p_control vs. norovirus_	0.609	1.0	**<0.01**	0.798	0.196	
**TAC** **(mmol/g protein)**	control	0.34 (0.32–0.38)	0.41 (0.38–0.46)	0.32 (0.31–0.36)	0.26 (0.22–0.27)	0.33 (0.32–0.44)	<0.001
	norovirus	**0.43** (0.41–0.53)	**0.16** (0.14–0.18)	**0.18** (0.11–0.24)	**0.25** (0.21–0.34)	**0.47** (0.44–0.48)	<0.001
	p_control vs. norovirus_	**<0.01**	**<0.01**	**<0.01**	0.256	0.070	
**TOS** **(IU/g protein)**	control	0.45 (0.29–0.53)	0.46 (0.43–0.56)	0.40 (0.38–0.46)	2.48 (2.20–2.76)	0.62 (0.60–0.71)	<0.001
	norovirus	**0.74** (0.67–0.90)	**0.55** (0.51–0.59)	**0.87** (0.86–1.07)	**1.97** (1.54–2.14)	**0.43** (0.40–0.46)	<0.001
	p_control vs. norovirus_	**<0.01**	0.097	**<0.01**	0.055	**<0.01**	
**MDA** **(IU/g protein)**	control	7.1 (6.5–8.2)	14.3 (13.5–15.1)	2.9 (2.8–3.6)	3.3 (3.1–3.8)	2.7 (2.1–3.2)	<0.001
	norovirus	**7.2** (6.7–8.4)	**12.1** (11.2–12.3)	**2.4** (2.1–2.6)	**3.7** (3.2–4.6)	**3.1** (2.8–3.8)	<0.001
	p_control vs. norovirus_	0.898	0.055	**<0.05**	0.523	0.246	

Abbreviations: CAT—catalase activity; GPx—glutathione peroxidase activity; GR—glutathione reductase activity; GST—glutathione S-transferase; MDA—malondialdehyde concentration; MnSOD—Mn superoxide dismutase activity; SOD—superoxide dismutase; TAC—total antioxidative capacity; TOS—total oxidative status. Green indicates biochemical analysis results showing an increase relative to the control. Biochemical analysis results showing a decrease relative to the control are highlighted in red. *p*—statistical significance (*p* ≤ 0.05).

## Data Availability

The data presented in this study are available on request from the corresponding author. The data are not publicly available due to the large number of results, which would affect the clarity of the manuscript.

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
