# Peer review of "Oxidative Stress Markers and Histopathological Changes in Selected Organs of Mice Infected with Murine Norovirus 1 (MNV-1)"

_ijms, 2024, doi:10.3390/ijms25073614_

Round 1
Reviewer 1 Report (Previous Reviewer 3)
Comments and Suggestions for Authors
My only suggestion is to include what the treatment groups are in the beginning of the results section so the reader doesn't need to flip through the manuscript to the materials and methods section to understand the results presented.
The revisions made improve the quality of the manuscript.
Author Response
Please see the attachement

Reviewer 2 Report (Previous Reviewer 2)
Comments and Suggestions for Authors
While the study is interesting, and I have no comments on the experimental design or results, it's necessary to rewrite the paper to achieve the writing quality required for publication in IJMS.
The Introduction is well-written, but I believe it could benefit from a rewrite to provide a clearer direction for the study. It's somewhat unclear whether the authors aim to delve deeper into the pathogenesis of MNV specifically or if they intend to explore oxidative stress and histopathologic changes in viral diseases more broadly, using MNV infection as a model. Rewriting the introduction to reflect this could enhance its effectiveness.
In the Results section, it's crucial to clearly specify the different groups at the outset rather than simply labeling them as Group 1, Group 2, etc. This ensures that readers have a clear understanding of these groups from the beginning. While I noticed this information is mentioned in the Materials and Methods section, it would be beneficial to reiterate it at the start of the Results section.
Improving the writing quality of the Results section is essential. Currently, the results seem to begin abruptly without proper reasoning or connection to the research objectives. Providing a more engaging introduction or establishing a clear connection to the authors' objectives would enhance readability for readers.
Sections 4.2 and 4.3 of the Materials and Methods appear to contain content that reads more like results rather than methods. While it's valuable to include this data, it may be more appropriate for the Results section rather than the Materials and Methods. Clarifying the distinction between these sections will improve the overall organization and clarity of the paper.
The Discussion section necessitates significant rewriting to improve engagement and coherence. It is crucial to emphasize core points and ensure a logical flow between them. Instead of presenting data randomly, it is imperative to articulate the key points the authors wish to convey and demonstrate their relevance to the study findings. For instance, in lines 196-211, an excessive amount of data is presented without focusing on the main point, which detracts from the discussion's clarity. It would be more effective to discuss the main points concisely, referring to previous studies and providing supporting data. Concerning lines 211-212, the statement "In our experiment, no changes were found in the histopathological studies, which seems very interesting because this fact was mentioned by other authors" appears contradictory to the authors' own results. Another instance is found in lines 266-275, where the authors dedicate an entire paragraph to external factors influencing oxidative stress markers; however, the significance of this in relation to the authors' study is scarcely discussed. Additionally, providing a strong introductory paragraph for the discussion will help set the stage and capture the readers' interest.
Author Response
Please see the attachement

Round 2
Reviewer 2 Report (Previous Reviewer 2)
Comments and Suggestions for Authors
The article has been improved after rewriting. I have only the following minor corrections:
1. The citation format for this article, given on the left side of the first page, has a slightly different title showing mice strain. Please correct it.
2. I noticed that in the Results section, the authors attempted to remove the term ‘groups’ and instead mentioned the number of days the mice were euthanized. However, the authors still mentioned ‘groups’ in lines 109 and 111. Please remove these terms and provide more clarity.
3. Perhaps the beginning of the Discussion can start with the general objective of the study, rather than beginning with why norovirus infection is not pathogenic in immunocompetent hosts. This can be the second paragraph. Starting the discussion with the objective of the study is more appropriate and interesting to readers.
4. Line 282: Check spelling.
5. Line 247: Spelling for 'lymphocyte'.
Author Response
Dear Reviewer,
Thank you very much for your comments, with which we fully agree.
We have marked all the corrections made in the text in blue :
- The citation format for this article, given on the left side of the first page, has a slightly different title showing mice strain. Please correct it.
Corrected
- I noticed that in the Results section, the authors attempted to remove the term ‘groups’ and instead mentioned the number of days the mice were euthanized. However, the authors still mentioned ‘groups’ in lines 109 and 111. Please remove these terms and provide more clarity.
Corrected
- Perhaps the beginning of the Discussion can start with the general objective of the study, rather than beginning with why norovirus infection is not pathogenic in immunocompetent hosts. This can be the second paragraph. Starting the discussion with the objective of the study is more appropriate and interesting to readers.
We have added an introductory sentence regarding the purpose of the study, which really improves the readability of the paragraph.
- Line 282: Check spelling.
Corrected
- Line 247: Spelling for 'lymphocyte'.
Corrected
Best regards
Barbara Bażanów
This manuscript is a resubmission of an earlier submission. The following is a list of the peer review reports and author responses from that submission.
Round 1
Reviewer 1 Report
Comments and Suggestions for Authors
The manuscript is poorly written and the results are not well argued in the discussion. Figure 4 is missing and figure captions should describe the groups in a more comprehensive way. In the results, the analyzed organs were liver, lung, kidney and brain, but in the figures spleen samples are shown.
I miss a discussion showing why it is possible the discrepancies between TOS and TAC results in some organs
Reviewer 2 Report
Comments and Suggestions for Authors
The manuscript by Janicka et al. describes changes in histopathology and oxidative stress markers in various tissues of mice during MNV infection. I find this study interesting, novel, and generally well-written. This type of study provides more insights into the pathogenesis of not only MNV infection but also various virus infections in humans. In my opinion, this study can be published without any further experiments. However, certain major and minor comments need to be addressed for this manuscript to be considered for publication.
Major Comments:
1. The viral strain used for this study is CW1. Why did the authors select this strain for the study, given that certain publications have reported differences in histologic changes with various viral strains? Mumphrey et al., 2007, reported that CW3 replicates more efficiently than CW1; thus, CW3 may exhibit a more pronounced difference than CW1.
2. Did the authors conduct any tests, such as PCR, to demonstrate that the mice were actually infected/equally infected with MNV after inoculation?
3. It is advisable to clearly specify what do groups 1, 2, 3, and 4 represent at the beginning of the results section.
4. In Figure 1D, authors showed a different region of brain compared to infected one. In my opinion, the same region should be shown for Group1.
5. What methodology was used for GST measurement? This information was not provided in the methodology section.
6. In which group (2, 3, or 4) were the biochemical assays performed, and what was the rationale for selecting this group? This information could be included in the paper for clarity.
7. The authors mentioned collecting small and large intestines, but the histopathologic changes in these tissues are not addressed in this study. Mumphrey et al., 2007, reported that MNV infection in immunocompetent mice resulted in mild inflammation of the intestine. Therefore, it would be beneficial to include observations on intestinal histopathology in this study for a more comprehensive understanding.
8. The authors have presented histopathological changes in the spleen, but oxidative stress markers were not assessed for this organ. Was there a specific reason for not testing oxidative stress markers in the spleen? Providing clarification on this aspect would enhance the completeness of the study.
9. The study distinctly demonstrates a notable difference in oxidative stress among various organs. Do the authors have any insights or comments on the potential reasons for this discrepancy, such as variations in viral replication between organs? Including these points in the discussion would provide valuable context for readers.
Minor Comments
1. The title mentions C56Bl/6J mice, while the manuscript refers to C57Bl/6J. Additionally, please check the standard way of writing. I assume the 'l' must be in uppercase (L).
2. I would recommend adding the full forms of abbreviations such as SOD, MnSOD, CAT, GR, and others in the abstract for clarity.
3. In the last paragraph of the introduction, the authors described that "The comprehension of relevant antioxidant mechanisms and the relationship between viral infection and oxidative stress may play a crucial role in understanding viral pathogenesis and identifying potential therapeutic targets, especially as antiviral therapies are often unavailable." How will this study, or similar studies, help identify potential therapeutic targets? Is it through reducing oxidative stress that can improve the clinical condition, despite not directly using antivirals? I would recommend slightly improving or elaborating on the paragraph from lines 94-100 to underscore the importance of this study.
4. The figures are currently categorized based on the mice groups. I would recommend classifying figures based on organ tissue. In other words, each figure could feature a control (Group 1) and infected tissues from Groups 2, 3, and 4. This is just a suggestion. Additionally, Figure 4 appears to be missing in the submitted version.
5. Panel labels are missing in Figure 3.
6. The scale in the figures is too small; please consider rearranging it for better visibility.
7. In Table 1, please consider increasing the font size of 'P control Vs norovirus'; it is currently difficult to read.
8. Perhaps I was not clear. What is the P value in the rightmost column of Table 1? If possible, please mention this in the legend for clarity.
9. In Table 1, I recommend highlighting the boxes that indicate an increase or decrease in values with different colors to aid comprehension for the reader. For example, use green for organs that show an increase compared to controls and red for a decrease.
10. The sentences from lines 156 to 167 could be combined and rephrased to improve overall readability.
11. I am unclear about the meaning of “(n=2)” mentioned in section 4.2.
12. It is just my opinion, and the authors/editors can decide on this, but I want to bring it to your attention. In the conflict of interest section, one of the authors declared their previous employment. However, if this does not directly or indirectly exert any financial or other types of influence on this study, then, in my opinion, there is no conflict of interest.
Reviewer 3 Report
Comments and Suggestions for Authors
This manuscript describes the effect of murine norovirus on the tissues and metabolites of C56BL/6J mice and will provide additional interesting information on oxidative stress in this mouse strain. However, the manuscript suffers from a lack of detail and comparison between past published results and the current study. Additionally, there is no description of why this particular mouse line was use rather than any other.
Specific comments:
L22: change "mouse" to "murine"
L23: change "used" to "uses"
L 31: delete "the authors suggest that"
L32: reword sentence. The viral infection results in changes to ROS not the virus itself
L47-L62: combine these 2 paragraphs and describe what causes immunosuppression.
L100 (and section 4.1 Mice): why are you using this particular strain of mice? What makes them different from all of the other strains used to study norovirus?
Figures 1, 2, and 3: It would be more useful to compare the same tissue at different treatments rather than to see multiple tissues together in each figure. Figure 3 doesn't contain A, B, C, nor D.
L130 lists a Figure 5 which does not exist at all in the manuscript.
Table 1 title is not consistent with the number or animals within each treatment listed in the Materials and Methods section. Are there more than one set of treatment groups? Please clarify and specify exactly what day post-infection is included on this table. It would also be helpful to bold the significant P values.
L143-L167: Please provide this information in paragraph form. Single sentences are not acceptable.
Discussion: The purpose of the discussion is to compare your results with that of part reports rather than to list those results in a separate paragraph.
L200: change "presented" to "present"
L221 -L238: this paragraph is missing citations
Section 4.1 Mice: again, why was this strain of mice chosen?
Section 4.2: there were only 2 mice used? This is unacceptable. As mentioned above, the title of Table 1 lists more mice. Please describe all experimental treatments and animal numbers used.
Section 4.4: how was the virus titred and stored?
L294: please list manufacturer's address
Section 4.6: Listing a reference for each of the assays is insufficient. A brief description of how the assays were performed including any changes made to the reference protocol needs to be included.
Round 2
Reviewer 1 Report
Comments and Suggestions for Authors
Even the manuscript has improved with all the corrections made by the authors, in my opinion, it has not enough quality to be published in this journal.